# Regression Tree Ensemble Rainfall–Runoff Forecasting Model and Its Application to Xiangxi River, China

**Aifeng Zhai [1], Guohua Fan [1], Xiaowen Ding [1,2,*] and Guohe Huang [2]**

[1] MOE Key Laboratory of Resources and Environmental System Optimization, College of Environmental Science and Engineering, North China Electric Power University, Beijing 102206, China; zhaiaifeng789@163.com (A.Z.); Guohua_Fan@hotmail.com (G.F.)

[2] Institute for Energy, Environment and Sustainable Communities, University of Regina, Regina, SK S4S 7H9, Canada; huang@iseis.org

[*] Correspondence: binger2000dxw@hotmail.com

**Abstract:** The development of an efficient and accurate hydrological forecasting model is essential for water management and flood control. In this study, the ensemble model was applied to predict the daily discharge; it not only could enhance the algorithm and improve the learning accuracy, but it was also the most effective representative model among various combinations of learning parameters. Using the survey data of Xingshan station in Xiangxi River, China, the suitability of the model was proven. The performance of the ensemble model was compared with the multiple linear regression model and the artificial neural network models. Furthermore, the length of the training samples and the peak value predictions were analyzed. The results showed that, firstly, the best effect of the discharge simulation model appeared in the ensemble model, while the simulation accuracy of the multiple linear regression model was lower than that of the artificial neural network model in some cases. Secondly, the prediction effect of the ensemble model for discharge was better than that of the single model to some extent, whereby the maximum absolute value of relative error was 8.11% using the ensemble model. A comprehensive analysis showed that the ensemble model was optimal. Furthermore, the ensemble model performed outstandingly in terms of hydrological forecasting. The ensemble model also provided theoretical support for hydrological forecasting and could be considered as an alternative to multiple linear regression models and artificial neural networks.

**Keywords:** hydrological forecasting; ensemble model; multiple linear regression; artificial neural network

## 1. Introduction

Rainfall–runoff (R–R) modeling plays a very important role in managing the activities of water resources such as flood control and reservoir operation [1,2]. However, the R–R system is extremely complex since it is influenced by weather, topography, underlying surface, and land usage [3,4], all of which have many kinds of temporal and spatial uncertainties [5]. Thus, creating and developing effective forecasting tools with high prediction accuracy is becoming urgent because of these complexities and uncertainties.

These forecasting tools can typically be divided into two main types: physical and data-driven models. Previously, many data-driven models were developed for mapping the R–R relations such as spatial autocorrelation (SAC) model [6], data-driven [7], linear regression [8], and artificial neural network models [9]. On the basis of the traditional hydraulic simulation model HEC-RAS, Kuriqi and Ardiclioglu [10] studied the hydraulic condition of the Loire. Kuriqi et al. [11] investigated the seepage process of Albania in different scenarios using numerical modeling. Compared with physical models, data-driven models can directly establish a mathematical relationship between the input and output data with a relatively simpler structure in operation. However, these models can hardly explicitly

describe the physical mechanism of hydrological processes. In addition, the accuracy of data-driven models still needs to be improved further according to different watersheds.

In recent years, as an important technology in data-driven models, the data-mining technique has been used in hydraulics and hydrology. Najafzadeh and Niazmardi [12] proposed a support vector regression model and applied it to biochemical oxygen demand (BOD) and chemical oxygen demand (COD) estimation in water. Najafzadeh et al. [13] predicted the water quality index in Karun River, Iran, using four common data-driven models. Moreover, a single data-driven model may not achieve the required prediction accuracy. Ensemble models have been proven to be effective forecast tools, whereby various results of multiple weaker models are integrated with certain rules to obtain a better forecast [14,15]. The ensemble model was introduced and applied to R–R modeling in [16–18], and its performance was generally better than each individual model in terms of simulation accuracy and generalization ability [19]. The regression tree ensemble (RTE), also referred to as a forest or simply an ensemble, is a tried-and-true technique for reducing the error of single machine-learned models. By learning multiple models over different subsamples of data and taking a majority vote at prediction time, the risk of overfitting a single model to all the data is mitigated. Popular ensemble models include Bayes modal averaging (BMA), bagging, boosting, and random forest [20–22]. The advantages of RTE include (1) the building or development of a binary tree through the selection of a splitting variable and recursively splitting of the data into two exclusive branches or nodes, (2) pruning to reduce the size of the tree until the optimum tree size is achieved, and (3) assigning a predictive value at each terminal branch.

Therefore, the objective of this study was to apply an RTE model to R–R modeling of Xiangxi River, China. To be specific, the RTE was developed on the basis of 3 year hydro-climatic data at Xingshan station in Xiangxi River. To calibrate and verify the parameters of the developed models, different combinations of factors affecting runoff were tested. Furthermore, this paper discusses the balance between structure and performance of the RTE model. Furthermore, the optimal effectiveness of RTE is demonstrated through a comparison of the results with multiple linear regression and artificial neural networks. This study also provides an improved hydrological forecasting model for water management and flood control.

## 2. Materials and Methods

To ensure the clarity of the methodology in this paper, a flowchart is shown in Figure 1. Using the collected data of daily rainfall and water level, multiple linear regression models and artificial neural networks were compared with the RTE with respect to hydrological forecasting. The performance of the models in terms of effect factors, daily runoff, and peak runoff was evaluated in order to demonstrate the superiority of the RTE.

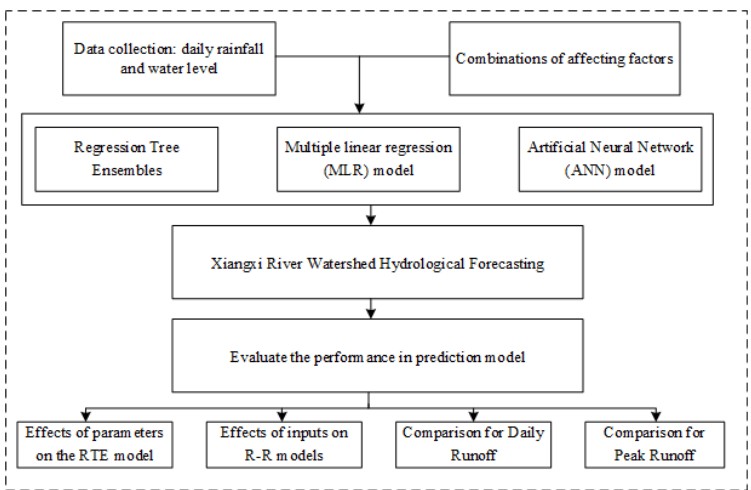

**Figure 1.** Flowchart of methodology.

### 2.1. Regression Tree Ensemble (RTE)

The ensemble model involves melding results from many weak learners into one high-quality ensemble predictor [23]. As a very successful predictive modeling approach, different variants of tree ensembles have been explored and used extensively (such as bagging and random forests). The subsections below outline the advantages of using tree-based methods and tree ensembles, as well as present the obstacles that need to be overcome for their successful application to hydrological prediction [24].

In analysis, a decision tree can be used to visually and explicitly represent decisions and decision making. In data mining, a decision tree describes data but not decisions. On the contrary, the resulting classification tree can be used as input for decision making. The goal of decision tree learning is to create a model that predicts the value of a target variable on the basis of several input variables.

Each interior node corresponds to one of the input variables, whereas edges represent each of the possible values of that input variable. Each leaf represents a value of the target variable given the values of the input variables represented by the path from the root to the leaf. Each element of the domain of the classification is called a class. A decision tree or a classification tree is a tree in which each internal (non-leaf) node is labeled with an input feature. The arcs coming from a node labeled with a feature are labeled with each of the possible values of the feature. Each leaf of the tree is labeled with a class or a probability distribution over the classes.

Regularization is a process of choosing fewer weak learners for an ensemble in a way that does not diminish predictive performance, which can regularize regression ensembles and regularize a discriminant analysis classifier in a non-ensemble context. The regularization method finds an optimal set of learner weights with minimum $\alpha_t$.

$$\sum_{n=1}^{N} w_n g\left(\left(\sum_{t=1}^{T} \alpha_t h_t(x_n)\right), y_n\right) + \lambda \sum_{t=1}^{T} |\alpha_t|, \tag{1}$$

where $\lambda$ is the lasso parameter, and $h_t$ is a weak learner in the ensemble trained on $N$ observations with predictors $x_n$, responses $y_n$, and weights $w_n$. The squared error of Equation (1) is presented below.

$$g(f, y) = (f - y)^2. \tag{2}$$

The ensemble is regularized on the same $(x_n, y_n, w_n)$ data used for training; thus,

$$\sum_{n=1}^{N} w_n g\left(\left(\sum_{t=1}^{T} \alpha_t h_t(x_n)\right), y_n\right). \tag{3}$$

Equation (3) is the ensemble re-substitution error. The error is measured by the mean squared error (MSE). The main procedure for R–R modeling using the RTE was as follows: firstly, prepare the response data and input the predictor data in a matrix; the data resolution used in this study was daily according to the collected data samples. Secondly, set the number of ensemble members after choosing an applicable ensemble method, and then prepare the weak learners. Lastly, a suitable ensemble can be obtained. For different combinations of input variable parameters (i.e., evaporation and flow several days in advance were taken out as independent variables), the optimum fitting results were selected as predictors. Fitting of 2 year calibration data and 3 year validation data were required. The response data were the two runoff datasets measured. In this model, the set parameter of a randomly selected tree was 100, and the default tree selection was used.

### 2.2. Multiple Linear Regression (MLR) Model

Regression analysis is one of the earliest applications and most widely used methods in long-term hydrological forecasting [25]. This method is the most common and basic one

in the statistical analysis of climate hydrological current [26]. In a hydrological system, a variable is often influenced by many other factors [3]. Therefore, the impact of multiple predictors must be taken into consideration in forecasting objects, as it is insufficient to use only one predictor for long-term hydrological forecasting. Accordingly, multiple regression analysis can be applied as an appropriate tool to solve these problems. Regression analysis is also a statistical method to study the causal relationship between two or more variables [27], which is useful for quantitative analysis and forecasting techniques.

Define $Y$ as the response variable (i.e., discharge) and $X_1, X_2, \ldots, X_p$ as the ensemble of predictor variables [28], where $p$ indicates their number (i.e., daily rainfall, daily evaporation, daily maximum temperature, daily minimum temperature, and daily discharge). The relationship between $Y$ and $X_1, X_2, \ldots, X_p$ can be represented by the following regression formula:

$$Y = \beta_0 + \beta_1 X_1 + \cdots\cdots + \beta_p X_p + \varepsilon,\tag{4}$$

where $\varepsilon$ is the random error, which is the representative of the approximate difference, and $\beta$ is the regression coefficient (constant). The function $f(X_1, X_2, \ldots, X_p)$ describes both $Y$ and $X_1, X_2, \ldots, X_p$ and the relationship between different datasets. The matrix of Equation (4) is expressed as

$$Y = \begin{bmatrix} y_1 \\ y_2 \\ \vdots \\ y_n \end{bmatrix}, \ X = \begin{bmatrix} 1 & x_{11} & \cdots & x_{p1} \\ 1 & x_{12} & \cdots & x_{p2} \\ \vdots & \vdots & & \vdots \\ 1 & x_{1n} & & x_{pn} \end{bmatrix}, \ \beta = \begin{bmatrix} \beta_1 \\ \beta_2 \\ \vdots \\ \beta_p \end{bmatrix}, \ \xi = \begin{bmatrix} \varepsilon_1 \\ \varepsilon_2 \\ \vdots \\ \varepsilon_n \end{bmatrix}\tag{5}$$

Analysis of variance (ANOVA) was used to characterize Equation (4). The null hypothesis was $\beta_j = 0$, where $j = 1, 2, \ldots, p$. The alternative hypothesis was that $\beta_j \neq 0$.

$$F = \frac{SS_r/p}{SS_e/(n-p-1)} \sim F(p, n-p-1),\tag{6}$$

$$\begin{aligned} SS_r &= \sum_{i=1}^{n} (\hat{y}_i - \bar{y})^2, \\ SS_e &= \sum_{i=1}^{n} (y_i - \hat{y}_i)^2 \end{aligned}\tag{7}$$

where $SS_r$ is the residual sum of squares, and $SS_e$ is the explained sum of squares. The equations obey the F distribution of freedom degree $(p, n-p-1)$.

### 2.3. Artificial Neural Network (ANN) Model

An ANN is a special computational model whose development was inspired by some biological features including run elements (neurons), as well as training and recall algorithms [29,30]. In almost all situations, an ANN is a self-adaptive system whose structure can be changed to optimize parameters in the learning phase.

Many studies have achieved system identification and modeling using neural network models [31]. The output of the model has two parts: the weighted sum of inputs and the introduction of different bias terms delivered to the level of activation by means of a transfer function.

The unit makes arrangements for a feedforward neural network with a hierarchical feed forward topology [32]. Such networks consist of input, hidden, and output layers. The input layer contains all input factors such as the daily rainfall, daily evaporation, daily maximum temperature, daily minimum temperature, and daily discharge. There can be several hidden layers featuring different types of neurons. Therefore, the number of hidden layers and neurons, representing the network structure, can be adjusted to improve network performance [33]. The decision on the number of neurons used in the hidden layer usually depends on the arithmetical mean of the number of inputs and outputs, while the arithmetic mean value of the input and output usually determines their use in decision

making. In the back propagation (BP) network used herein, the number of nodes in the input layer was 5 and that in the output layer was 1. Generally, the optimal number of layers in a network is 3 or 4. In this study, a three-layer network structure was used as the network model, and five factors were selected as input layer neurons. The five input variables were used to establish the training sample of the neural network prediction model.

*2.4. Performance Indices*

The performance of the prediction model was evaluated using four valuation criteria: Judge coefficient ($R^2$) [34], Nash efficiency coefficient (*NE*) [35], the root-mean-square error (RMSE) [36], and the mean absolute percentage error (MAPE) [37]. The indices can be calculated as follows:

$$R^2 = \left[ \frac{\sum\limits_{i=1}^{n} (Q_i^0 - \overline{Q}^0)(Q_i^m - \overline{Q}^m)}{\sqrt{(Q_i^0 - \overline{Q}^0)(Q_i^m - \overline{Q}^m)}} \right], \tag{8}$$

$$NE = 1 - \frac{\sum\limits_{i=1}^{n} (Q_i^0 - Q_i^m)^2}{\sum\limits_{i=1}^{n} (Q_i^0 - \overline{Q}^0)^2}, \tag{9}$$

$$RMSE = \sqrt{\frac{1}{n} \sum\limits_{i=1}^{n} (Q_i^0 - Q_i^m)^2}, \tag{10}$$

$$MAPE = \frac{1}{n} \sum\limits_{i=1}^{n} \left| \frac{Q_i^0 - Q_i^m}{Q_i^0} \right| \times 100, \tag{11}$$

where $Q_i^0$ is the observed discharge at moment *i*, $Q_i^m$ is the predicted discharge at moment *i*, $\overline{Q}^0$ is the average discharge of the observed values, and $\overline{Q}^m$ is the average discharge of the predicted values. $R^2$ indicates the correlation quality of between the predicted values and observed values. *NE* is often used to evaluate the predictive ability of a hydrological model. $R^2$ and *NE* values closer to 1 denote a more accurate model. The root-mean-square error evaluates the residuals between the predicted and observed values. MAPE is the weighted average of the absolute error. Smaller RMSE and MAPE values denote a more accurate model.

**3. Case Study**

*3.1. Study Area*

In this study, Xiangxi River in Xingshan County was selected as the target area (Figure 2). Xiangxi River is the first middling tributary of the Chang Jiang River near the Three Gorges Dam. After the Three Gorges Reservoir was constructed, some reservoir bays were built, with Xiangxi Bay being a typical representative. Xingshan County covers an area of 2327 km$^2$; it is located in the western Hubei Province of China and near the Yangtze River. Its climate is subtropical continental monsoon, with an annual average temperature of 15.3 °C, annual average solar radiation of 99,000 card/cm$^2$, annual precipitation of 900–1200 mm, and average annual precipitation of 134 mm per day. Furthermore, rain is abundant in Xingshan in the summer, with 41% of the total precipitation. To guarantee the accuracy of the calculations and validate the model, this paper collected meteorological and hydrological data from Xingshan Station (110°25′–111°06′ E, 31°04′–31°34′ N). The dataset contained the information of daily rainfall and water level for the period of 3 years (1991 to 1993), of which 2 years (from 1991 to 1992, 731 data) were used for calibration and 1 year (1993, 365 data) was used for validation (Table 1).

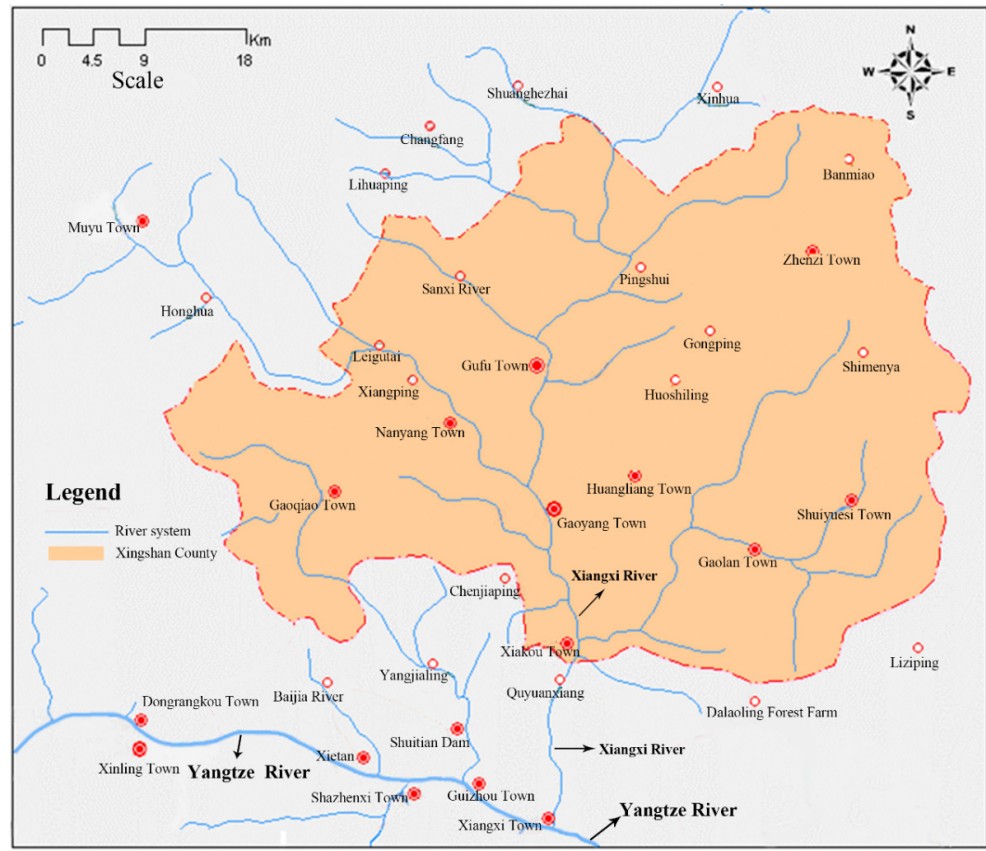

**Figure 2.** Diagram of the study area.

**Table 1.** Parameters in Xiangxi catchment.

| Statistical Parameters | Daily Precipitation | | Daily Evaporation | | Daily Discharge | | Daily High Temperature | | Daily Low Temperature | |
|---|---|---|---|---|---|---|---|---|---|---|
| | 1991–1992 | 1993 | 1991–1992 | 1993 | 1991–1992 | 1993 | 1991–1992 | 1993 | 1991–1992 | 1993 |
| Maximum | 119.9 | 81.8 | 12 | 12 | 684 | 525 | 41.6 | 40.1 | 27.3 | 26.7 |
| Minimum | 0 | 0 | 0 | 0 | 7.99 | 8.75 | 0.6 | 2.7 | −6.9 | −2.8 |
| Average | 2.484 | 2.793 | 3.591 | 3.021 | 33.129 | 39.144 | 22.785 | 21.951 | 12.545 | 12.351 |
| Standard deviation | 7.903 | 7.801 | 2.617 | 2.362 | 57.747 | 48.490 | 9.076 | 8.919 | 7.645 | 7.585 |

*3.2. Hydrological Forecasting for Xiangxi River Watershed*

The RTE, MLR, and ANN were developed using a combination of factors including daily rainfall, daily evaporation, daily maximum temperature, daily minimum temperature, and daily discharge. Five representative combinations of factors were employed to evaluate their effects on the RTE, MLR, and ANN models. These models for future runoff forecasting were based on hydrological data using different factors and different days in advance. In the process of prediction, the optimal number of days in advance was determined at first for each factor. Then, to ensure the accuracy of the simulation results, five different combinations of factors were constructed. Accordingly, these models could be described using the following formulas:

$$Q_{t1} = \text{model}(P_{t-4}, E_{t-4}, H_{t-4}, L_{t-4}, F_{t-4}), \tag{12}$$

$$Q_{t2} = \text{model}(P_{t-1}, E_{t-4}, E_{t-8}, L_{t-8}, F_{t-1}, F_{t-2}), \tag{13}$$

$$Q_{t3} = \text{model}(P_{t-1}, F_{t-1}, F_{t-2}, F_{t-10}), \tag{14}$$

$$Q_{t4} = \text{model}(F_{t-1}, F_{t-2}, F_{t-9}, F_{t-10}), \tag{15}$$

$$Q_{t5} = \text{model}(P_{t-1}, P_{t-8}), \tag{16}$$

where $Q_t$ is the daily average flow rate on the prediction day, $t$ is the timepoint representing the day of interest, $t - i$ ($i = 1$–10) represents the different days of each factor in advance, $P$ is precipitation, $E$ is evaporation, $H$ is the daily maximum temperature, $L$ is the daily minimum temperature, and $F$ is the discharge. The performance of different combinations of factors can be seen in Table 2.

**Table 2.** Performance of different combinations of parameters.

| Days in Advance | Daily Precipitation | Daily Evaporation | Daily Max Temperature | Daily Min Temperature | Daily Discharge |
|---|---|---|---|---|---|
| 1 | 0.6363 | 0.4617 | 0.6136 | 0.7239 | 0.8179 |
| 2 | 0.4023 | 0.6429 | 0.6355 | 0.6972 | 0.7031 |
| 3 | 0.4334 | 0.6551 | 0.6702 | 0.6459 | 0.6417 |
| 4 | 0.4120 | 0.7099 | 0.6572 | 0.6520 | 0.6654 |
| 5 | 0.3833 | 0.6929 | 0.5896 | 0.6261 | 0.6378 |
| 6 | 0.2968 | 0.6156 | 0.6430 | 0.7014 | 0.6069 |
| 7 | 0.4349 | 0.5853 | 0.6143 | 0.6652 | 0.6348 |
| 8 | 0.5257 | 0.7120 | 0.6215 | 0.7838 | 0.6557 |
| 9 | 0.4513 | 0.6944 | 0.5908 | 0.6674 | 0.6716 |
| 10 | 0.2847 | 0.5730 | 0.6954 | 0.6659 | 0.6994 |

## 4. Results Analysis

### 4.1. Comparison of Models

The RTE was applied for R–R modeling. The candidate factors ($X$) were 1 day in advance, 2 days in advance, 4 days in advance, etc. Specifically, the rainfall, evaporation, and discharge several days in advance were taken as the independent variables. Fitting results were obtained using this program with a training set and prediction set, before choosing the best one as the predictive factor. Therefore, the first 2 years of data were used for calibration and the third year data were used for validation. For this model, an ensemble of 100 trees was randomly chosen using the default tree options. Specifying a regression tree using surrogate splits allows improving the predictive accuracy in the presence of NaN values. Finally, the regression tree ensemble was trained using the function and 100 learning cycles.

In this study, the ensemble model was compared with MLR and ANN models trained with different combinations of statistical parameters and multistep forecasting applications with different rainfall–runoff characteristics. As shown in Table 3, the model calibration and verification showed that the simulated and observed values were correlated, thus improving the $R^2$ coefficient and RMSE. The comparison of all three models in modeling future discharge revealed that the ensemble model provided a better fit. According to Table 3, the following conclusions can be drawn:

(1) The prediction accuracy and generalization ability were significantly improved compared to the single model and the network in an ideal state, indicating that the ensemble model established for discharge forecasting is feasible and effective. The ensemble model integrated the advantages of each single model, effectively avoiding the errors of the single model being too large and having unstable defects. It had the characteristics of high-precision forecasting, strong generalization ability, and error smoothening.

(2) According to the predicted results from the comparison of each single model, the prediction accuracy of the ANN model was better than that of the MLR model. However, according to the fitting results of the training samples, the fitting effect of the MLR model was equivalent to that of the ANN model. Furthermore, according

to the forecast values of the test samples, the generalization ability of both the MLR model and the ANN model was poor.

(3) As a whole, as a single model, the absolute value of the average relative error of prediction was less than 15% for both the MLR and the ANN model, and the absolute value of the maximum relative error was less than 29.55%, which can meet the precision requirement of discharge forecasting to some extent. However, their accuracy was inferior to that of the RTE.

**Table 3.** Performance in predicting daily discharge for Xiangxi catchment.

| Models | | Calibration | | | | Verification | | | |
|---|---|---|---|---|---|---|---|---|---|
| | | *Z* | *R*$^2$ | *NE* | **RMSE** | *Z* | *R*$^2$ | *NE* | **RMSE** |
| Five factors | RTE | 0.6293 | 0.5562 | 0.5963 | 42.30 | 0.6028 | 0.3650 | 0.3450 | 40.76 |
| | MLR | 0.5893 | 0.5210 | 0.4580 | 48.90 | 0.5536 | 0.2463 | 0.2891 | 41.25 |
| | ANN | 0.5900 | 0.5223 | 0.4230 | 46.65 | 0.5645 | 0.2866 | 0.2923 | 41.02 |
| Four factors | RTE | 0.7273 | 0.6752 | 0.6725 | 32.96 | 0.6146 | 0.3777 | 0.3624 | 38.36 |
| | MLR | 0.6608 | 0.4367 | 0.4325 | 43.40 | 0.5272 | 0.2780 | 0.2576 | 41.32 |
| | ANN | 0.6721 | 0.5036 | 0.4420 | 40.27 | 0.5341 | 0.2853 | 0.2840 | 41.11 |
| Two factors | RTE | 0.7096 | 0.5031 | 0.5035 | 40.75 | 0.5429 | 0.2948 | 0.2894 | 40.83 |
| | MLR | 0.6560 | 0.4303 | 0.4303 | 43.65 | 0.5229 | 0.2734 | 0.2704 | 41.45 |
| | ANN | 0.6691 | 0.4829 | 0.4351 | 40.30 | 0.5070 | 0.2571 | 0.2500 | 41.91 |
| Daily precipitation | RTE | 0.6106 | 0.3728 | 0.3724 | 45.80 | 0.5569 | 0.3102 | 0.2931 | 40.38 |
| | MLR | 0.5273 | 0.2780 | 0.2780 | 49.14 | 0.5590 | 0.3125 | 0.2983 | 43.32 |
| | ANN | 0.5368 | 0.2974 | 0.2891 | 46.29 | 0.5260 | 0.2767 | 0.2860 | 41.35 |
| Daily discharge | RTE | 0.6936 | 0.4811 | 0.4807 | 41.65 | 0.5709 | 0.3259 | 0.3343 | 39.92 |
| | MLR | 0.6263 | 0.3922 | 0.3299 | 45.08 | 0.5710 | 0.3261 | 0.3213 | 39.92 |
| | ANN | 0.6201 | 0.4065 | 0.3091 | 45.28 | 0.5716 | 0.3269 | 0.2967 | 39.90 |

### 4.2. Comparison of Daily Runoff

In this study, the applications of RTE, MLR, and ANN models revealed different characteristics in hydrological forecasting. Compared with MLR, ANN was more suitable for rain-based hydrological forecasts. Neural networks can provide high accuracy, which is consistent with the conclusions of most results reported in the literature [38]. In this study, the results using four factors in the second group were the best among all factor combinations, as further discussed below.

Figure 3 provides a verification and comparison of the three prediction models, where the red lines represent the measured values. As shown in the figure, the tendency of the ensemble model in predicting the results followed the real data trends, and the error was minimum. The average absolute value of the relative error of discharge forecasting was 5.67% using the ensemble model for the Xingshan hydrological station of Xiangxi River. The maximum absolute value of relative error was 8.11%. The trend of the ensemble model validation prediction results was similar to the actual data, and the prediction results were more stable. On the other hand, the trend of the MLR and ANN prediction results deviated in some places, while the daily forecast of MLR was better than that of ANN. However, when the timescale was changed to months or years, the prediction advantage of MLR was not obvious. In terms of performance indices, the ensemble model had higher accuracy in the case of daily discharge data prediction.

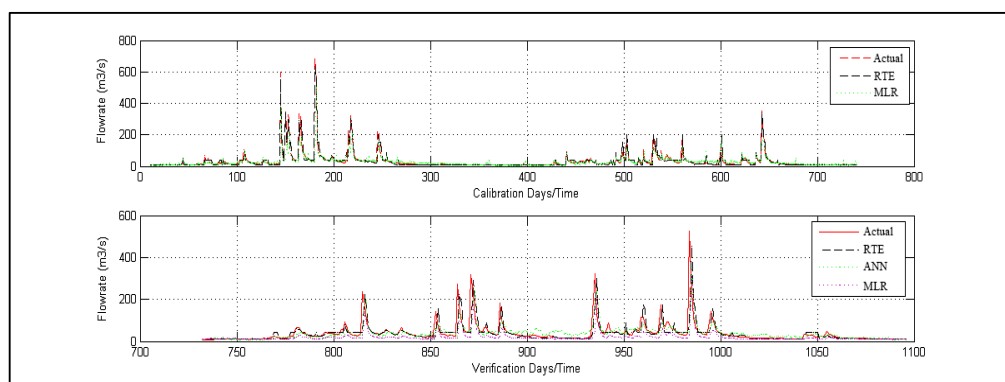

**Figure 3.** Verification and comparison of three prediction models.

*4.3. Comparison of Peak Runoff*

Figure 4 provides a comparison of the measured values and the predicted values of maximum peak using the three models. The serial numbers represent the order in which peaks occurred. The red symbols represent the 10 maximum peaks of measured values, revealing a similar peak trend prediction to the three models; however, it is clear that the black symbols representing the peak of the ensemble model had the best corresponding relationship. The ensemble model could achieve performance that is comparable to MLR and ANN, improving the precision and generalization ability of discharge forecasting.

The plot reveals that the prediction results of the ensemble model were more accurate. Because of the complexity of the Xiangxi River system and the uncertainty of variation, the prediction effect of the regression forecasting method based on statistical theory was not ideal in this area, although it could reflect the change trend of the data sequence. However, the fluctuations in the data were often too large, causing errors in the regression prediction model at extreme points. Therefore, the application of the regression method to predict discharge needs to be studied further. Due to the ANN's inherent defects of slow convergence rate and ease of falling into a local minimum, it is largely restricted in discharge forecasting applications. Thus, improving prediction accuracy and generalization ability has important practical significance and application value for hydrological prediction.

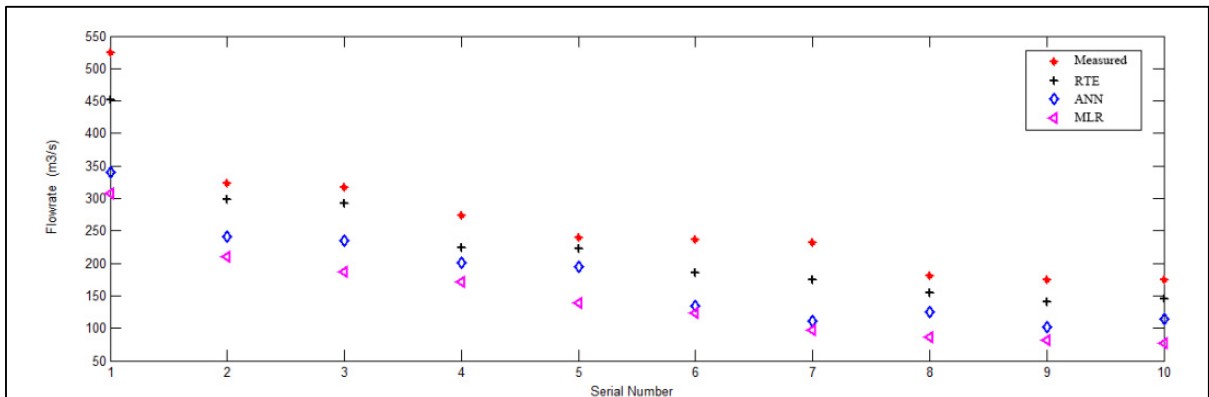

**Figure 4.** Comparison of measured values and the predicted values of maximum peak using the three models.

**5. Conclusions**

A regression tree ensemble was applied to daily discharge forecasting. Xiangxi River was used to demonstrate the applicability of the ensemble model through a comparison with MLR and artificial neural network (ANN) models. The results showed the viability of this model in obtaining comparable performance with a shorter training time. Other problems such as BP models with different input times, ensemble model capacity, training

sample length, and long-term peak prediction were also analyzed. We can conclude that the performance is sensitive to the number of inputs and length of the training data.

This paper introduced the application of an ensemble model to hydrological forecasting and compared various prediction schemes with MLR and ANN models. Considering that the MLR computing time is highly sensitive to the number of inputs, numerous and precise data should be gathered for simulations using MLR [39]. In this study, the result showed that the MLR daily forecast was better than that of the ANN due to the large amount of data collected. However, compared with the ANN, the MLR lost its advantage for monthly and annual forecasting. The results of this study are consistent with the literature [40,41]. The ensemble model showed the features of high accuracy, strong generalization ability, and stable error change, making full use of the comprehensive advantages of each model. In the discharge forecasting of Xiangxi River, the results showed that the maximum absolute value of relative error was 8.11% when using the ensemble model. Similar to the findings by Kim et al. [42], the streamflow prediction results of this study also showed that the ensemble model had a higher forecasting accuracy than individual models. Thus, the ensemble model represents an alternative with high accuracy for researchers and engineers engaged in hydrologic forecasting. In the future, further parameterization can be introduced to improve the model's performance [43]. Moreover, the performance of the ensemble model can be investigated in river basins of different scale, as well as for multistep prediction.

**Author Contributions:** Conceptualization, A.Z. and G.H.; methodology, A.Z., X.D. and G.H.; software, A.Z.; validation, A.Z. and G.F.; formal analysis, A.Z.; investigation, G.F.; resources, X.D.; data curation, A.Z.; writing—original draft preparation, A.Z.; writing—review and editing, A.Z., G.F. and X.D.; visualization, A.Z.; supervision, X.D. and G.H.; project administration, X.D.; funding acquisition, X.D. and G.H. All authors have read and agreed to the published version of the manuscript.

**Funding:** This research work was funded by the National Key Scientific and Technological Projects of the PRC (2014ZX07104-005) and the Fundamental Research Funds for the Central Universities of the PRC (2015XS103). Furthermore, this research work was funded by the National Key R&D Program of China CERC-WET Project (2018YFE0196000), the National Key Research and Devel-opment Program of China (2017YFC0404503), and the National Natural Science Foundation of China (41601529).

**Institutional Review Board Statement:** Not applicable.

**Informed Consent Statement:** Not applicable.

**Data Availability Statement:** This paper collected meteorological and hydrological data from Xingshan Station, Hubei Province of China.

**Acknowledgments:** This research work was funded by the National Key Scientific and Technological Projects of the PRC (2014ZX07104-005) and the Fundamental Research Funds for the Central Universities of the PRC (2015XS103). Furthermore, this research work was funded by the National Key R&D Program of China CERC-WET Project (2018YFE0196000), the National Key Research and Development Program of China (2017YFC0404503), and the National Natural Science Foundation of China (41601529). The authors gratefully acknowledge the financial support of the programs and agencies.

**Conflicts of Interest:** The authors declare no conflict of interest.

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
