# Peer review of "Regression Tree Ensemble Rainfall–Runoff Forecasting Model and Its Application to Xiangxi River, China"

_water, doi:10.3390/w14030463_

Round 1
Reviewer 1 Report
The current manuscript tested the efficiency of using the Regression Tree Ensemble model in rainfall-runoff forecasting compared to two other models (MLR and ANN) in Xiangxi River. Although the topic is vitally important for water management and floods control, the current manuscript lacks a comprehensive discussion of its main findings. Authors need to essentially improve the results and discussion part/s to turn their findings into practical knowledge that helps other researchers understand the main problem and suggested solutions such as the best inputs and model parameters for accurate estimations. Also, the language is another major aspect that needs to be largely improved, particularly the grammatical errors. Some other detailed suggestions are listed below:
- Line 10, Correspondence word is repeated
- In the title, Xiangxi “River” instead of Xiangxi.
- Line 50, introduced
- Line 71, Methodology and Methods, do you mean Materials & Methods?
- Line 173, section 3. Case study has only one subsection, “3.1. Study Area”. So, the section could be 2.1. Study area and can be the first subsection of 2. Materials & methods
- The first row of Table 1 is Maximum or minimum? Please check
- In column 1 of Table 2, what do you mean by factor? Do you mean days in advance? If so, please give it the correct name
- Section 4.1 describes a method of how the random forest model was trained and validated rather than focusing on results analysis.
- Figure 3 is not readable enough. Its size and quality need to be improved. The name of the axis can be “actual discharge” or “measured discharge” on the Y-axis and “predicted” or “modeled” discharge on X-axis.
- The R-squared and RMSE are missing from Figure 3 and must be added.
- Section 4.1. Effects of parameters on the RTE model, what do you mean by parameters? The authors mentioned they had used 100 trees with default tree options (lines 219-220). So no various parameters were tested. If the authors mean different numbers and days in advance of each factor, the section needs to include the results of those other trials.
- Section 4.2 Effects of inputs is actually reporting the estimation accuracy of the three models rather than discussing the effect of different inputs in Table 3. Moreover, the text describes the average absolute value of the relative error of discharge, but Table 3 has RMSE rather than absolute error values.
- Figure 5, the x-axis is “serial number” of what? Stations?
- Line 305, “MLR would lose its advantage in the monthly and annual forecasting”. It is not clear which part of the results tested this finding? Same in line 314, “higher parameterization would make the model present better performance [41]”. This was not discussed before
Author Response
On behalf of my co-authors, we thank you very much for giving us an opportunity to revise our manuscript, we appreciate editor and reviewers very much for their positive and constructive comments and suggestions on our manuscript entitled “Regression Tree Ensemble Rainfall-Runoff Forecasting Model and its Application in the Xiangxi River, China” (ID: water-1554474). Those comments are all valuable and very helpful for revising and improving our manuscript, as well as the important guiding significance to our researches. We have studied comments carefully and have made correction which we hope meet with approval. We have tried our best to revise our manuscript according to the comments in the revised manuscript. We would like to express our great appreciation to you and reviewers for comments on our manuscript. The main corrections in the manuscript and the responds to the reviewer’s comments have been shown in the response letter and the revised manuscript.

Reviewer 2 Report
This research investigates Regression Tree technique (as a data-mining model) to predict the rainfall-runoff and consequently new insight into the Xiangxi river. The paper falls within the scope of special issue and scientific materials of the present draft is of interest for readers of water journal. However, the paper requires moderate revisions before it is proceeded:
(1) Quantitative performance of RTE model can be added to the abstract section as a sound comparison.
(2) In the "Multiple linear regression (MLR) model" section, authors need to present this equations through ANOVA.
(3) Setting parameters of RTE and ANN need to be discussed in the paper.
(4) Eq.(5) has been incorrectly written. This equation needs refinements.
(5) Selection of Eqs.(9)-(12) needs criteria. Authors are recommended to use justifications.
(6) Scatter plots of Fig.3 needs major modifications. X and Y axis of scatter plots need to become at the same length.
(7) Recently, there are so many data-mining techniques which were used in the hydraulics and hydrology. Introduction section can be furnished by the following literature:
A Novel Multiple-Kernel Support Vector Regression Algorithm for Estimation of Water Quality Parameters
Reliability assessment of water quality index based on guidelines of national sanitation foundation in natural streams: integration of remote sensing and data-driven models
Author Response

(The authors gave the same response as above.)

Round 2
Reviewer 1 Report
Thanks to the authors for considering all my earlier comments. No more comments from my side
Best Regards
Author Response
On behalf of my co-authors, we thank you very much for giving us an opportunity to revise our manuscript, we appreciate you very much for your positive and constructive comments and suggestions on our manuscript entitled “Regression Tree Ensemble Rainfall-Runoff Forecasting Model and its Application in the Xiangxi River, China” (ID: water-1554474). Those comments are valuable and very helpful for our manuscript, as well as the important guiding significance to our researches.

Reviewer 2 Report
Authors have not addressed comments 6 and 7.
Author Response
On behalf of my co-authors, we thank you very much for giving us an opportunity to revise our manuscript, we appreciate editor and reviewers very much for their positive and constructive comments and suggestions on our manuscript entitled “Regression Tree Ensemble Rainfall-Runoff Forecasting Model and its Application in the Xiangxi River, China” (ID: water-1554474). Those comments are all valuable and very helpful for revising and improving our manuscript, as well as the important guiding significance to our researches. We have studied comments carefully and have made correction which we hope meet with approval. We have tried our best to revise our manuscript according to the comments 6 and 7 in the revised manuscript. We would like to express our great appreciation to you and reviewers for comments on our manuscript.

This manuscript is a resubmission of an earlier submission. The following is a list of the peer review reports and author responses from that submission.